# SSD-GS: Scattering and Shadow Decomposition for Relightable 3D Gaussian Splatting

**Iris Zheng, Guojun Tang, Alexander Doronin, Paul Teal, Fang-Lue Zhang**[*]
Victoria University of Wellington

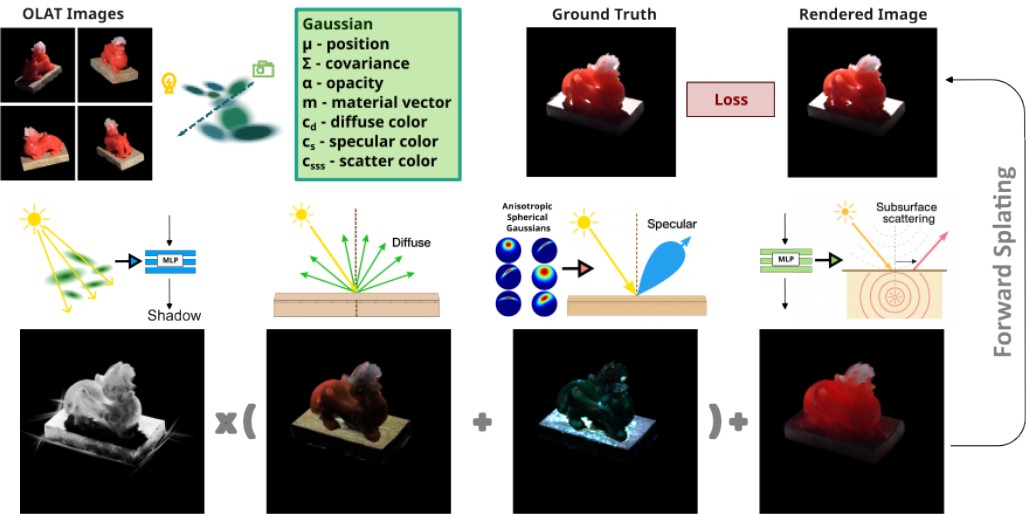

Figure 1: Overview of the proposed SSD-GS pipeline. Our method incorporates four physically inspired reflectance terms: diffuse, specular, shadow, and subsurface scattering, to model realistic light–material interactions. These components are progressively introduced during training, allowing the network to gradually disentangle complex illumination effects and improve relighting fidelity under unseen lighting conditions.

## Abstract

We present SSD-GS, a physically-based relighting framework built upon 3D Gaussian Splatting (3DGS) that achieves high-quality reconstruction and photorealistic relighting under novel lighting conditions. In physically-based relighting, accurately modeling light-material interactions is essential for faithful appearance reproduction. However, existing 3DGS-based relighting methods adopt coarse shading decompositions, either modeling only diffuse and specular reflections or relying on neural networks to approximate shadows and scattering. This leads to limited fidelity and poor physical interpretability, particularly for anisotropic metals and translucent materials. To address these limitations, SSD-GS decomposes reflectance into four components: diffuse, specular, shadow, and subsurface scattering. We introduce a learnable dipole-based scattering module for subsurface transport, an occlusion-aware shadow formulation that integrates visibility estimates with a refinement network, and an enhanced specular component with an anisotropic Fresnel-based model. Through progressive integration of all components during training, SSD-GS effectively disentangles lighting and material properties, even for unseen illumination conditions, as demonstrated on the challenging OLAT dataset. Experiments demonstrate superior quantitative and perceptual relighting quality compared to prior methods and pave the way for downstream tasks, including controllable light source editing and interactive scene relighting. The source code is available at: https://github.com/irisfreesiri/SSD-GS.

---

[*]Fang-Lue Zhang (fanglue.zhang@vuw.ac.nz) is the corresponding author

# 1 INTRODUCTION

Photorealistic 3D reconstruction with relightable capabilities has become increasingly important across domains such as AR/VR for digital humans, cinematic visual effects, cultural heritage preservation, and medical simulation. Traditional methods (Levoy & Hanrahan, 1996; Seitz & Dyer, 1996; 1997; Snavely et al., 2006), however, typically compromise either geometric precision or photorealistic quality, particularly in complex lighting conditions or with reflective and textured surfaces. While these approaches enabled view synthesis under captured illumination, they relied on explicit geometric reconstructions and provided no means to disentangle reflectance from lighting. As a result, they cannot support relighting under novel illumination, which is essential for realistic appearance reproduction in many applications. More recently, neural rendering approaches, in particular those based on Neural Radiance Fields (NeRF) (Mildenhall et al., 2020), have made notable progress by jointly encoding geometry and appearance in an implicit volumetric representation. Methods such as DNL (Gao et al., 2020) and NRHints (Zeng et al., 2023) introduce explicit lighting supervision and learnable shading representations to support relightable view synthesis. However, NeRF-based methods typically suffer from high computational cost, which limits their practicality for interactive or real-time applications.

3D Gaussian Splatting (3DGS), initially developed for real-time radiance field rendering, has emerged as a compelling alternative to NeRF-style implicit representations that rely on ray marching, offering superior computational efficiency and rendering fidelity. Recent extensions of 3DGS for relightable rendering fall into two main paradigms. Some methods assume static lighting conditions during training (Jiang et al., 2023; Liang et al., 2024; Chen et al., 2024; Gao et al., 2024), which fundamentally lacks their flexibility for photorealistic relighting. Others leverage dynamic lighting configurations such as one-light-at-a-time (OLAT) capture setups (Bi et al., 2024; Kuang et al., 2024; Fan et al., 2025; Dihlmann et al., 2025), offering more physically plausible supervision but making it difficult to disentangle material properties from illumination. This disentanglement is crucial for simulating complex light transport behaviors of real-world materials, where nonlinear interactions give rise to visually critical phenomena such as gradient soft shadows and subsurface scattering. Consequently, developing robust techniques to model these intricate lighting-material interactions remains a substantial technical challenge for relightable 3D reconstruction.

We propose **SSD-GS**, a physically-based relighting method designed for 3DGS, where "physically-based" follows the real-time PBR convention and denotes the use of physically inspired reflectance models within an efficient rasterized framework. Built upon the 3DGS pipeline, our framework explicitly decomposes complex reflectance into four components: diffuse, specular, subsurface scattering, and shadow. Our main contributions are:

- We introduce a learnable dipole-based scattering module that simulates realistic subsurface scattering effects using physically motivated diffusion profiles.
- We design an occlusion-aware shadow formulation that combines a visibility prior with a learned refinement network, enabling accurate modeling of view- and light-dependent shadowing effects.
- We progressively integrate all reflectance components (diffuse, specular, shadow, and subsurface scattering) during training and refine both lighting and camera conditions, leading to improved relighting quality and stronger generalization under novel illuminations.

# 2 RELATED WORKS

Accurate relighting and novel view synthesis require recovering both scene geometry and material appearance under illumination. We review NeRF- and 3DGS-based relighting methods, followed by subsurface scattering (SSS) models for physically plausible rendering.

**NeRF-based Relighting.** Neural Radiance Fields (NeRF)(Mildenhall et al., 2020) represent scenes as volumetric fields optimized from posed RGB images, enabling photorealistic novel view synthesis under fixed lighting. Extensions for relighting factorize appearance into reflectance and illumination using priors or explicit transport modeling. For instance, NeRV(Srinivasan et al., 2021), NeRD (Boss et al., 2021), and NeRFactor (Zhang et al., 2021b) disentangle reflectance under static lighting with geometry-aware priors, while PhySG (Zhang et al., 2021a) uses spherical Gaussians to represent

BRDFs and environment lighting. To address directional lighting, ReNeRF (Xu et al., 2023) models near-field OLAT illumination via a spherical codebook and light transport decoder, enabling spatially varying lighting. NRHints(Zeng et al., 2023) injects OLAT-derived shadow and highlight hints into a NeRF-style radiance field, achieving relighting effects comparable to DNL(Gao et al., 2020) but using a fully volumetric, single-branch design. However, NeRF-based methods suffer from implicit, non-physical representations, making decomposition hard to interpret or control. Moreover, they are computationally expensive, requiring hours of training per scene.

**Gaussian-based Relighting.** 3DGS (Kerbl et al., 2023) models scenes as point clouds of anisotropic Gaussians with learned extent, opacity, and view-dependent appearance. While efficient for view synthesis, its SH-based color encoding (Ramamoorthi & Hanrahan, 2001; Sloan et al., 2002) is inherently limited to smooth, low-frequency angular variations, which reduces expressiveness for capturing high-frequency effects such as specular highlights and scattering. Several extensions enhance 3DGS with physically motivated components, including GaussianShader (Jiang et al., 2023), GI-GS (Chen et al., 2024), and R3DG (Gao et al., 2024). However, these typically assume static lighting conditions, which prevents them from generalizing to novel illuminations. Their relightable variants usually perform global relighting using environmental maps, but lack the ability to model precise changes in individual light sources. To overcome these limitations, recent works exploit dynamic lighting conditions, most notably one-light-at-a-time (OLAT) datasets. $GS^3$ (Bi et al., 2024) decomposes reflectance by modeling diffuse and specular terms at the Gaussian level, while handling shadows and other residual effects at the pixel level in a deferred rendering style (Ye et al., 2024). However, this design struggles to capture complex light transport phenomena such as soft shadows and indirect illumination. OLAT Gaussians (Kuang et al., 2024) use directional encodings with two MLPs to model incident and scattering components, but their use of a proxy mesh for normal supervision makes them highly sensitive to the quality of the proxy geometry. RNG (Fan et al., 2025) achieves improved shadow quality by introducing a latent appearance code, which replaces physically meaningful shading representations and thus sacrifices interpretability. Inspired by these OLAT-based approaches, we introduce physically interpretable shading to better disentangle lighting–material interactions and extend performance to more diverse datasets.

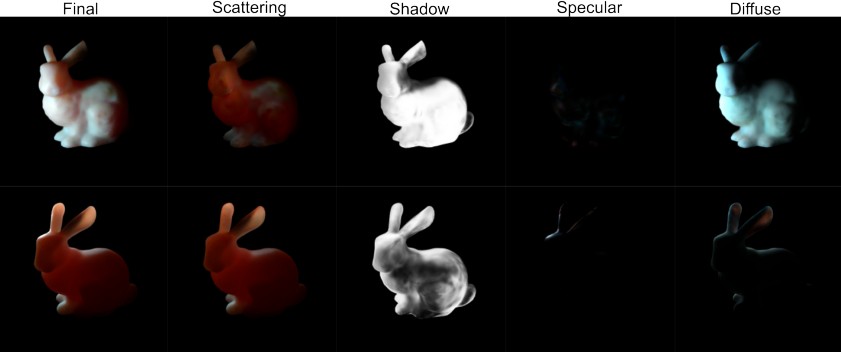

Figure 2: Relighting results from our SSD-GS pipeline. The same *Bunny* view under two different lighting conditions from the SSS-GS synthetic dataset (Dihlmann et al., 2025).

**Subsurface Scattering.** Subsurface scattering (SSS) has been extensively studied for simulating light transport in translucent materials such as skin, jade, wax, and marble. Classical approaches, including the standard dipole (Jensen et al., 2001), quantized dipole (D'Eon & Irving, 2011), and directional dipole (Frisvad et al., 2014), offer efficient and realistic approximations. Extensions such as shape-adaptive dipole models (Vicini et al., 2019) and advanced BSSRDF formulations (Yan et al., 2017) further enhance accuracy and generality. More recently, subsurface scattering has been explored in neural rendering frameworks through learning-based techniques. Neural SSS (Tg et al., 2024) approximates the translucent appearance using per-view and per-light neural reflectance fields, but it relies heavily on dense supervision and lacks physical interpretability. In the context of Gaussian Splatting, SSS-GS (Dihlmann et al., 2025) directly learns the subsurface scattering radiance via a neural network conditioned on Gaussian and lighting inputs. The output is blended with BRDF shading using a learned weight, treating SSS as a residual term rather than a physically motivated subsurface model. In contrast, we integrate a physically grounded subsurface scattering approach

into the 3DGS pipeline, based on the standard dipole diffusion approximation (Jensen et al., 2001). This classical method provides a closed-form BSSRDF that approximates multiple scattering in homogeneous media. By embedding it into the Gaussian Splatting framework, we enable efficient, interpretable simulation of soft scattering effects, while maintaining modular compatibility with other shading components such as diffuse, specular, and shadow terms.

## 3 PRELIMINARY

Our method builds on the 3D Gaussian Splatting (3DGS) framework (Kerbl et al., 2023), which represents a scene as a set of anisotropic 3D Gaussians. Each Gaussian is defined by its center $\mathbf{x}_i$, opacity $\alpha_i$, and a covariance matrix $\Sigma_i$. The covariance matrix is parameterized via a rotation matrix $R_i$ and a scaling matrix $S_i$, such that $\Sigma_i = R_i S_i S_i^\top R_i^\top$. During rendering, the Gaussians are projected onto the image plane and composited using front-to-back alpha blending as:

$$\mathbf{C}_{\text{pixel}} = \sum_{i=1}^{N} T_i \cdot \alpha_i \cdot \mathbf{C}_i, \quad T_i = \prod_{j=1}^{i-1}(1 - \alpha_j) \tag{1}$$

Each Gaussian color $\mathbf{C}_i$ is computed using a view-dependent SH expansion:

$$\mathbf{C}_i(\mathbf{v}) = \sum_{b=1}^{B} \mathbf{c}_{i,b} \cdot Y_b(\mathbf{v}) \tag{2}$$

While effective for encoding smooth appearance, this SH-based model lacks physical grounding and struggles to capture high-frequency view-dependent effects. In this work, we replace it with a decomposed physically-based model to better capture full-frequency light–material interactions.

## 4 METHODOLOGY

We extend the 3D Gaussian Splatting (3DGS) framework by incorporating a physically-based reflectance model that replaces its original spherical harmonics (SH)-based appearance representation. Our formulation decomposes shading into four components—diffuse, specular, shadow, and subsurface scattering (SSS)—each modeled either analytically or using lightweight neural fields. These components are evaluated per-Gaussian and composited to form the final image, enabling interpretable supervision and relightable rendering under novel illumination. Their visual effects are illustrated in Fig. 2, and a detailed ablation study is provided in Sec. 5.3. An overview of the formulation is illustrated in Fig. 1.

### 4.1 PBR-BASED SHADING

We formulate a physically-based shading function that operates directly on the 3D Gaussian representation. Unlike prior work that employs view-dependent spherical harmonics (SH) for color synthesis (Kerbl et al., 2023), we decompose reflectance into physically interpretable terms, enabling improved photorealism, per-component supervision, and controllable relighting. The color of each Gaussian is computed as:

$$\mathbf{C}_i = (c_d f_d + c_s f_s) \cdot S(\mathbf{x}) + c_{sss} f_{sss} \tag{3}$$

where: $f_d, f_s, f_{sss}$ denote the scalar reflectance intensities for diffuse, specular, and subsurface scattering, defined in Eqs. 9, 10, and 5, respectively; $c_d, c_s, c_{sss} \in \mathbb{R}^3$ are the corresponding learned base colors for each reflectance term; $S(\mathbf{x})$ denotes the soft shadow factor, computed as a density-weighted average over shadow rays and further refined using an MLP, with its detailed formulation given in Eq. 8. This decomposition is evaluated per Gaussian and composited through the 3DGS forward-rendering pipeline, where alpha blending accumulates Gaussian contributions into the final pixel color. The resulting image is supervised with a pixel-wise loss against the ground truth. A detailed analysis of the interaction between the shadow term and subsurface scattering is provided in Appendix E.2.

## 4.2 Subsurface Scattering Term

We model subsurface scattering (SSS) using the standard dipole diffusion profile, with scattering properties defined per Gaussian. To predict these parameters, we train a neural field $\Theta_{\text{SSS}}$ that maps spatial and directional inputs to the corresponding scattering coefficients.

$$\{\sigma_s, \sigma_a, r\} = \Theta_{\text{SSS}}(\mathbf{x} \mid \mathbf{w}_o, \mathbf{w}_i, \mathbf{n}, \mathbf{m}) \tag{4}$$

where $\mathbf{x}$ denotes the Gaussian center, $\omega_i$ and $\omega_o$ are the light and view directions, $\mathbf{n}$ is the surface normal derived from its local z-axis, and $\mathbf{m} \in \mathbb{R}^6$ is a learnable per-Gaussian material embedding; $\sigma_s, \sigma_a$, and $r$ denote the scattering coefficient, absorption coefficient, and surface separation distance used in the dipole formulation.

The subsurface scattering (SSS) predictor is implemented as a 6-layer MLP with a hidden size of 256 and ReLU activations. It takes as input the positional encodings (with $L = 4$ frequency bands) of the spatial location $\mathbf{x}$, the viewing direction $\omega_o$, and the lighting direction $\omega_i$, as well as the local surface normal $\mathbf{n}$ and a per-Gaussian material embedding $\mathbf{m}$. To ensure physical plausibility and improve training stability, the network outputs are passed through sigmoid activations and rescaled to fall within plausible material-specific ranges: $\sigma_s, \sigma_a \in [0.05, 2.05]$, and $r \in [0.1, 3.1]$. We evaluate the standard dipole diffusion profile (Jensen et al., 2001) as:

$$f_{sss}(r) = \frac{\alpha'}{4\pi} \left( z_r(\sigma_t d_r + 1)\frac{e^{-\sigma_t d_r}}{d_r^3} + z_r z_v(\sigma_t d_r + 1)\frac{e^{-\sigma_t d_v}}{d_v^3} \right) \tag{5}$$

where $\alpha' = \frac{\sigma_s}{\sigma_s + \sigma_a}$, $\sigma_t = \sigma_s + \sigma_a$, $z_r$ and $z_v$ are the depths of the real and virtual dipole sources, determined by the optical parameters $(\sigma_s, \sigma_a, \eta)$, and $d_r, d_v$ are the corresponding distances from the shading point to the real and virtual dipole sources, computed from the surface separation $r$.

Our SSS formulation combines physically grounded modeling with learnable parameter prediction, enabling realistic reproduction of subsurface scattering effects without requiring external geometry (Kuang et al., 2024). Because surface normals and material properties are inferred directly from the Gaussian representation, the system remains robust under challenging geometric conditions. As a result, it generalizes well to complex or noisy regions where mesh-derived normals may be unreliable, thus preserving effective scattering estimation.

## 4.3 Shadow Term

We model soft shadows using a two-stage approach that combines per-ray shadow evaluation with neural refinement. In the first stage, for each Gaussian we trace a shadow ray from the light source to every pixel and accumulate transmittance into visibility cues. In the second stage, a compact neural module takes these cues, together with geometry and material features, and predicts a scalar decay factor used in shading.

**Stage 1: Shadow Evaluation.** Given a light direction $\omega_i$, each Gaussian considers the set of pixels $i$ covered by its 2D projection. For each pixel, we evaluate a shadow ray from the light source toward that pixel and accumulate the opacity of intervening Gaussians. This yields a continuous per-ray transmittance

$$v_i = \prod_{k \in \mathcal{O}_i} (1 - \alpha_k), \tag{6}$$

where $\mathcal{O}_i$ is the depth-ordered set of Gaussians intersected by the shadow ray, and $\alpha_k \in [0, 1]$ denotes the opacity of Gaussian $k$.

To obtain a soft shadow estimate, these per-ray transmittance values are aggregated using the Gaussian's projected density as weights. Let $\rho_i$ denote the projected density of Gaussian $g$ at pixel $i$. The coarse visibility of Gaussian $g$ is then defined as the density-weighted expectation,

$$\hat{v}_g = \frac{\sum_i \rho_i v_i}{\sum_i \rho_i}, \tag{7}$$

which summarizes how much light from the direction $\omega_i$ reaches the Gaussian $g$ after accounting for overlapping geometry, and serves as a compact visibility estimate.

**Stage 2: Neural Refinement.**   The coarse visibility $\hat{v}$ captures the primary directional shadowing trend but may miss fine variations arising from contact shadows, geometric details, and material-dependent attenuation. To account for these effects, we refine $\hat{v}$ using a lightweight neural module $\Theta_{\text{shad}}$, which predicts a shadow attenuation term as a function of position:

$$S(\mathbf{x}) = \Theta_{\text{shad}}(\mathbf{x} \mid \hat{v}, \omega_i, \mathbf{m}) \tag{8}$$

The shadow refinement network is implemented using a 3-layer MLP with 32 hidden units per layer and ReLU activations. Its inputs include the Gaussian center $\mathbf{x}$, incident light direction $\omega_i$, coarse shadow estimate $\hat{v}$, and material embedding $\mathbf{m}$. To capture high-frequency spatial and directional variation, both $\mathbf{x}$ and $\omega_i$ are encoded using positional encoding with $L = 3$ frequency levels.

The refined shadow term $S(\mathbf{x})$ modulates the diffuse and specular components of our shading model, while the scattering term is added separately. This produces the final illumination contribution for Gaussian $g$ as defined in Eq. 3.

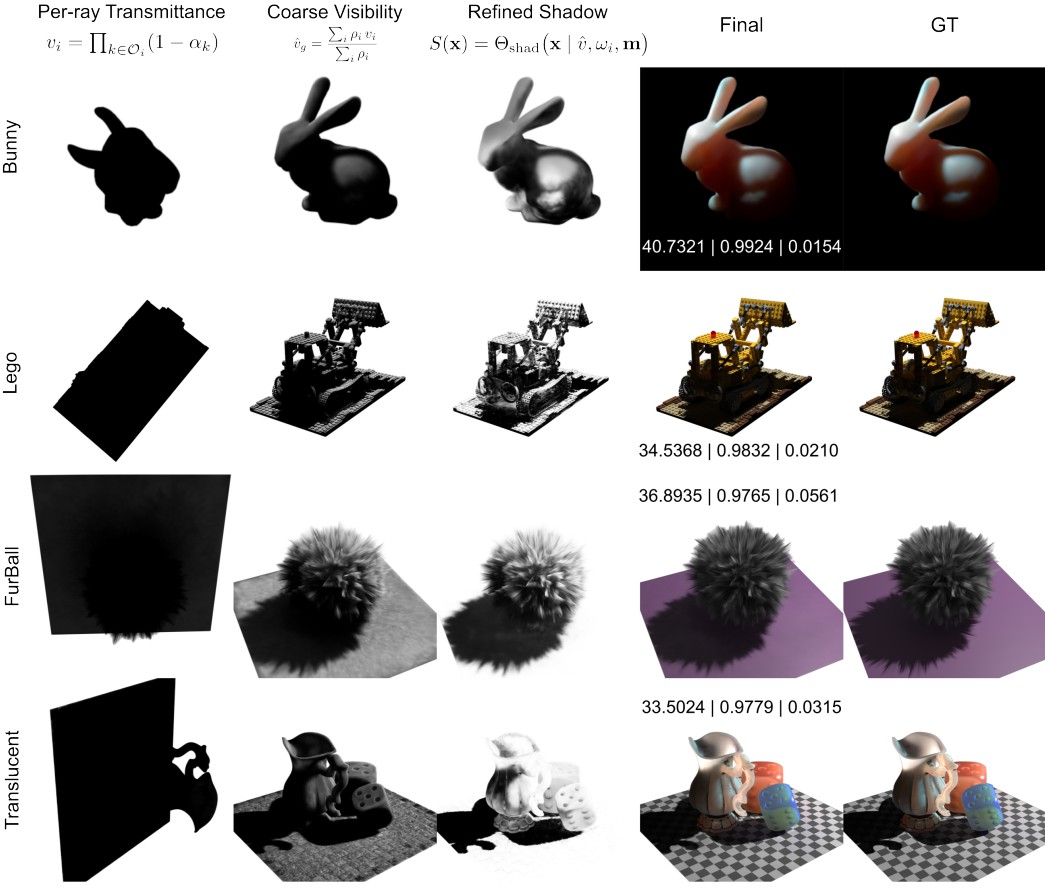

Figure 3: Shadow pipeline visualization. For each scene, we show per-ray transmittance $v_i$, coarse visibility $\hat{v}_g$, the refined shadow $S(x)$, and the final rendered result with metrics. The progression illustrates how continuous transmittance yields smooth, geometry-aware soft shadows.

A visual illustration of this progression from per-ray transmittance $v_i$, to coarse visibility $\hat{v}_g$, and finally to the refined shadow $S(\mathbf{x})$ is provided in Fig. 3. The figure highlights how continuous volumetric visibility naturally produces smooth, geometry-consistent soft shadows under point-light illumination. A complementary comparison against screen-space shadow accumulation methods is included in Appendix G.2.

### 4.4 DIFFUSE AND SPECULAR TERMS

We decompose direct shading into diffuse and specular components. The diffuse term is modeled with a Lambertian BRDF, which assumes uniform surface reflectance and produces view-independent, cosine-weighted reflection. Although simple, this model provides a stable foundation for capturing low-frequency appearance and ensures physically meaningful supervision during the early stages of training. The specular term, in contrast, accounts for high-frequency, view-dependent reflections. We represent it as a Fresnel factor (Schlick, 1994) modulated by anisotropic spherical Gaussian (ASG) bases (Xu et al., 2013). The Fresnel term captures the angular dependence of reflection intensity, particularly the sharp increase near grazing angles, while the ASG bases provide a compact yet expressive representation of anisotropic highlights. This formulation allows us to reproduce complex effects such as brushed metals and fabrics. Further technical details and equations are provided in Appendix B.

### 4.5 TRAINING METHODOLOGY

To stabilize convergence and reduce interference between reflectance components, we adopt a progressive training strategy. Four components are introduced in a coarse-to-fine order across simple phases defined by a small set of iteration thresholds (see Appendix A.1, Fig. 6 and Fig. 7 for details). A single default configuration is used for all scenes, ensuring that the approach remains stable and reproducible. Concurrently, we refine camera poses and lighting positions throughout training. The camera adjustment module is activated once the shadow term is introduced, while lighting position refinement begins during the specular phase. Experimental results are presented in our ablation study (see Sec. 5.3 and Appendix E.1).

## 5 EXPERIMENTS

We evaluate our relightable rendering method on both real-captured and synthetic OLAT datasets. This section first introduces the evaluated methods and datasets, followed by quantitative and qualitative comparisons. We then present ablation studies to assess the contribution of individual model components and training strategies. All experiments are conducted on a workstation equipped with an NVIDIA RTX 3090 GPU and an Intel Core i7-14700K CPU, running Windows 11 Education.

### 5.1 DATASETS

The OLAT datasets provide controlled illumination by sequentially activating individual point light sources, and are widely used benchmarks for evaluating relightable view synthesis. To ensure a consistent and challenging setup, test-time lighting directions are excluded from training.

**Real Dataset.** We use the seven OLAT-captured dataset provided by NRHints (Zeng et al., 2023): *Cat*, *CatSmall*, *CupFabric*, *Fish*, *FurScene*, *Pikachu*, and *Pixiu*. Each scene contains 500–1500 training images and 45–200 test views, all rendered against a black background. *CatSmall*, *CupFabric*, and *Pikachu* are rendered at a resolution of $1024\times1024$, while the remaining four use $512\times512$.

**Synthetic Datasets.** We use the six synthetic scenes released by $GS^3$ (Bi et al., 2024): *Translucent*, *AnisoMetal*, *Drums*, *FurBall*, *Hotdog*, and *Lego*. Each scene includes 2000 training images and 400 testing images at a resolution of $512\times512$, rendered against a white background. In addition, we evaluate on the five synthetic scenes from SSS-GS (Dihlmann et al., 2025): *Bunny*, *Candle*, *Dragon*, *Soap*, and *Statue*, which emphasize subsurface scattering effects. Each scene includes 500 training images and 500 test views at a downscaled resolution of $256\times256$ against a black background.

### 5.2 QUANTITATIVE AND QUALITATIVE ANALYSIS

First, we evaluate both reconstruction quality on the training set and relighting performance on the test set under unseen lighting conditions.

We then compare our method against four representative Gaussian Splatting–based approaches: vanilla 3DGS (baseline), GI-GS (Chen et al., 2024) as a representative of relighting under static

Table 1: Quantitative comparison results. The best/second-best results are colored in red / orange .

(a) Comparison with the original 3DGS (Kerbl et al., 2023), GI-GS (Chen et al., 2024), GS³ (Bi et al., 2024), and RNG (Fan et al., 2025) on the real datasets from NRHints (Zeng et al., 2023).

| Dataset | Cat | | CatSmall | | CupFabric | | Fish | | FurScene | | Pikachu | | Pixiu | |
|---|---|---|---|---|---|---|---|---|---|---|---|---|---|---|
| Method | Train | Test | Train | Test | Train | Test | Train | Test | Train | Test | Train | Test | Train | Test |
| **PSNR ↑** | | | | | | | | | | | | | | |
| 3DGS | 15.2225 | 14.5326 | 22.8367 | 22.5727 | 24.9219 | 25.0488 | 22.8247 | 22.8411 | 18.7746 | 18.4838 | 19.8235 | 19.6310 | 20.0114 | 18.5501 |
| GI-GS | 14.5256 | 13.9988 | 22.3667 | 22.3222 | 24.0188 | 24.3821 | 22.2452 | 22.7500 | 17.9882 | 17.8520 | 19.2010 | 19.1867 | 19.0030 | 18.1064 |
| GS³ | 30.0755 | 27.4081 | 34.8341 | 34.3136 | 36.5090 | 36.1375 | 31.5265 | 30.7218 | 28.6820 | 28.2228 | 30.0745 | 29.4128 | 30.6831 | 29.7001 |
| RNG | 27.7478 | 26.6059 | 34.7398 | 34.3709 | 37.7308 | 37.3219 | 29.1378 | 29.0835 | 27.9967 | 27.6930 | 31.6145 | 31.2646 | 29.8650 | 28.8554 |
| Ours | 30.0854 | 27.6844 | 35.2740 | 34.6472 | 38.0656 | 37.4702 | 32.0748 | 31.1646 | 31.7846 | 30.7349 | 32.4506 | 31.9298 | 33.6065 | 31.1213 |
| **SSIM ↑** | | | | | | | | | | | | | | |
| 3DGS | 0.7140 | 0.6962 | 0.9097 | 0.8896 | 0.9407 | 0.9430 | 0.8424 | 0.8312 | 0.7999 | 0.7869 | 0.9053 | 0.9000 | 0.8624 | 0.8298 |
| GI-GS | 0.3210 | 0.3162 | 0.8765 | 0.8750 | 0.9136 | 0.9178 | 0.7430 | 0.7437 | 0.5918 | 0.5811 | 0.8708 | 0.8724 | 0.6229 | 0.6117 |
| GS³ | 0.9240 | 0.9028 | 0.9777 | 0.9759 | 0.9825 | 0.9821 | 0.9306 | 0.9209 | 0.9426 | 0.9368 | 0.9621 | 0.9605 | 0.9457 | 0.9394 |
| RNG | 0.8556 | 0.8427 | 0.9709 | 0.9687 | 0.9803 | 0.9797 | 0.8909 | 0.8923 | 0.9195 | 0.9149 | 0.9673 | 0.9661 | 0.9244 | 0.9187 |
| Ours | 0.9224 | 0.9027 | 0.9786 | 0.9767 | 0.9839 | 0.9833 | 0.9363 | 0.9260 | 0.9576 | 0.9518 | 0.9675 | 0.9637 | 0.9524 | 0.9452 |
| **LPIPS ↓** | | | | | | | | | | | | | | |
| 3DGS | 0.2983 | 0.3033 | 0.1149 | 0.1183 | 0.0894 | 0.0868 | 0.1700 | 0.1814 | 0.2044 | 0.2101 | 0.1116 | 0.1103 | 0.1473 | 0.1721 |
| GI-GS | 0.3496 | 0.3518 | 0.1310 | 0.1339 | 0.1151 | 0.1116 | 0.1996 | 0.2063 | 0.2460 | 0.2492 | 0.1371 | 0.1347 | 0.2957 | 0.3089 |
| GS³ | 0.1228 | 0.1338 | 0.0624 | 0.0659 | 0.0501 | 0.0506 | 0.0836 | 0.0910 | 0.0778 | 0.0807 | 0.0711 | 0.0717 | 0.0795 | 0.0826 |
| RNG | 0.1959 | 0.2023 | 0.0586 | 0.0619 | 0.0373 | 0.0376 | 0.1269 | 0.1292 | 0.1084 | 0.1105 | 0.0507 | 0.0514 | 0.1021 | 0.1057 |
| Ours | 0.1251 | 0.1357 | 0.0621 | 0.0656 | 0.0503 | 0.0506 | 0.0779 | 0.0855 | 0.0690 | 0.0724 | 0.0679 | 0.0679 | 0.0751 | 0.0791 |

(b) Comparison with 3DGS (Kerbl et al., 2023), GI-GS (Chen et al., 2024), GS³ (Bi et al., 2024), and RNG (Fan et al., 2025) on the GS³ synthetic datasets.

| Dataset | Translucent | | AnisoMetal | | Drums | | FurBall | | Hotdog | | Lego | |
|---|---|---|---|---|---|---|---|---|---|---|---|---|
| Method | Train | Test | Train | Test | Train | Test | Train | Test | Train | Test | Train | Test |
| **PSNR ↑** | | | | | | | | | | | | |
| 3DGS | 17.1853 | 16.4899 | 18.1692 | 17.1009 | 26.5180 | 24.5093 | 21.5473 | 20.1206 | 19.3050 | 16.9535 | 19.0612 | 15.9886 |
| GI-GS | 17.1222 | 16.0766 | 17.7309 | 15.9567 | 26.7554 | 24.6177 | 21.3335 | 19.5295 | 19.1535 | 16.8118 | 19.5919 | 16.4229 |
| GS³ | 31.1327 | 32.1999 | 30.1878 | 28.8219 | 34.0111 | 33.2688 | 34.6201 | 34.9845 | 32.1779 | 32.7244 | 31.2224 | 30.5617 |
| RNG | 28.1919 | 28.5659 | 26.4611 | 25.9203 | 20.4970 | 20.3033 | 24.5084 | 23.4342 | 29.4095 | 29.5277 | 18.5810 | 18.4872 |
| Ours | 32.6058 | 32.3919 | 31.1077 | 30.0448 | 34.2448 | 33.5514 | 35.4793 | 35.1639 | 32.4901 | 32.1330 | 31.1434 | 30.4664 |
| **SSIM ↑** | | | | | | | | | | | | |
| 3DGS | 0.8984 | 0.8958 | 0.8995 | 0.8849 | 0.9556 | 0.9439 | 0.9095 | 0.8951 | 0.8956 | 0.8599 | 0.8514 | 0.7904 |
| GI-GS | 0.8651 | 0.8586 | 0.8537 | 0.8304 | 0.9066 | 0.8941 | 0.8720 | 0.8592 | 0.8636 | 0.8282 | 0.8128 | 0.7647 |
| GS³ | 0.9787 | 0.9775 | 0.9702 | 0.9635 | 0.9865 | 0.9841 | 0.9747 | 0.9707 | 0.9764 | 0.9745 | 0.9704 | 0.9581 |
| RNG | 0.9586 | 0.9598 | 0.9440 | 0.9393 | 0.9199 | 0.9244 | 0.9277 | 0.9204 | 0.9608 | 0.9572 | 0.8756 | 0.8616 |
| Ours | 0.9835 | 0.9823 | 0.9762 | 0.9698 | 0.9870 | 0.9848 | 0.9776 | 0.9733 | 0.9776 | 0.9743 | 0.9706 | 0.9570 |
| **LPIPS ↓** | | | | | | | | | | | | |
| 3DGS | 0.0755 | 0.0748 | 0.0638 | 0.0704 | 0.0371 | 0.0442 | 0.0918 | 0.0865 | 0.0882 | 0.1128 | 0.1101 | 0.1416 |
| GI-GS | 0.1137 | 0.1155 | 0.1084 | 0.1179 | 0.1142 | 0.1242 | 0.1643 | 0.1694 | 0.1368 | 0.1638 | 0.1458 | 0.1643 |
| GS³ | 0.0247 | 0.0254 | 0.0304 | 0.0341 | 0.0145 | 0.0160 | 0.0566 | 0.0524 | 0.0297 | 0.0305 | 0.0323 | 0.0401 |
| RNG | 0.0438 | 0.0402 | 0.0490 | 0.0491 | 0.0691 | 0.0685 | 0.1290 | 0.1264 | 0.0473 | 0.0484 | 0.1419 | 0.1470 |
| Ours | 0.0201 | 0.0200 | 0.0255 | 0.0295 | 0.0144 | 0.0157 | 0.0482 | 0.0442 | 0.0293 | 0.0326 | 0.0331 | 0.0419 |

(c) Comparison with SSS-GS (Dihlmann et al., 2025) and KiloOSF (Yu et al., 2022) on the SSS-GS synthetic datasets. For baselines, we report the average results directly from the respective papers, while the per-scene results of our method are provided in the Tab. 3

| Dataset | Train (Average) | | | Test (Average) | | | Other Metrics | | |
|---|---|---|---|---|---|---|---|---|---|
| Method | **PSNR ↑** | **SSIM ↑** | **LPIPS ↓** | **PSNR ↑** | **SSIM ↑** | **LPIPS ↓** | **FPS** | **Train T.** | **GPU** |
| KiloOSF | - | - | - | $25.91 \pm 1.88$ | $0.93 \pm 0.02$ | $0.097 \pm 0.03$ | 14.4 | > 20 h | RTX 4090 |
| SSS-GS | - | - | - | $35.01 \pm 1.01$ | $0.972 \pm 0.01$ | $0.040 \pm 0.01$ | $154.8 \pm 28.26$ | < 1 h | RTX 4090 |
| Ours (w/o Opt) | 40.7087 | 0.9907 | 0.0123 | 37.4409 | 0.9843 | 0.0186 | $66.28 \pm 14.37$ | < 2 h | RTX 3090 |
| Ours (w/ Opt) | 41.8705 | 0.9924 | 0.0099 | 38.3542 | 0.9863 | 0.0158 | $61.50 \pm 16.23$ | ≈ 2.5 h | RTX 3090 |

illumination, and GS³ (Bi et al., 2024) and RNG (Fan et al., 2025) as representatives of OLAT-based relighting. All methods are trained for 100K iterations with identical settings, and experiments are conducted on both the NRHints real dataset and the GS³ synthetic dataset for fair comparison.

Finally, to further validate the effectiveness of our physically based SSS shading term, we compare against SSS-GS (Dihlmann et al., 2025) and KiloOSF (Yu et al., 2022) on the SSS-GS synthetic dataset, using the quantitative results reported in the SSS-GS paper. Following the experimental setup in (Dihlmann et al., 2025), our method is trained for 60K iterations and rendered on a black background to ensure comparability.

**Quantitative Results.**   As shown in Tab. 1, our method achieves consistently strong performance across both training and test sets. By introducing a physically based decomposition of shading terms, our approach yields clear numerical advantages on datasets with pronounced scattering and specular effects, while achieving comparable results to other relighting methods on datasets dominated by low-frequency appearance, demonstrating strong generalization across diverse scenarios.

**Qualitative Results.**   Fig. 4 presents visual comparisons on the real-world scenes, while additional results on synthetic datasets are provided in Appendix C.1 and C.2. Compared to existing

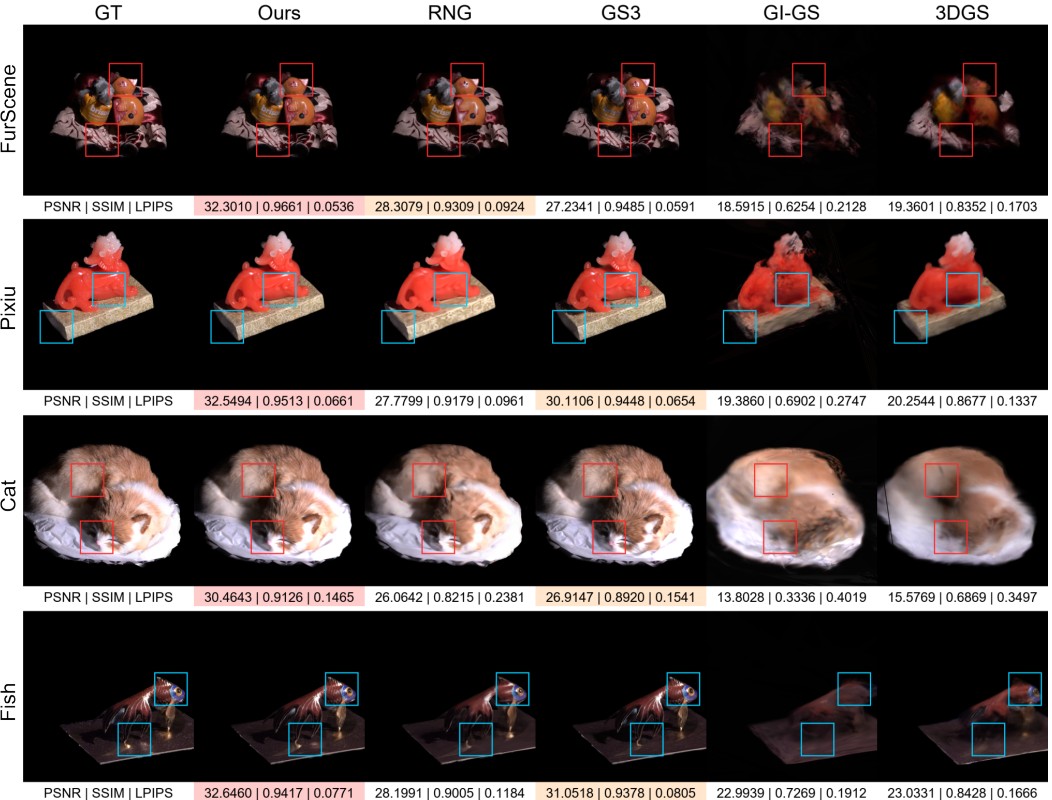

Figure 4: Qualitative comparison on real datasets from NRHints (Zeng et al., 2023). It presents relighting results on the test set under novel lighting. The best/second-best results (based on PSNR) are highlighted in red / orange .

approaches, our method produces relighting results that are consistently more faithful to the ground truth, especially in challenging scenes with complex material properties and light–material interactions. In particular, GS$^3$ often fails to capture sharp shadow boundaries and tends to introduce noise in shadow regions, notably in scenes such as *Fish* and *FurBall*. RNG, while capturing reasonable global appearance, frequently loses fine-scale reflectance and geometry details. For instance, the cat's nose is reconstructed as a flat white region instead of retaining its pink tone and curvature, and specular floor textures in the *Fish* scene are oversmoothed under strong lighting. These qualitative differences demonstrate our model's ability to preserve both soft shading and high-frequency details and its robust generalization to unseen lighting.

## 5.3 ABLATION STUDY

**Reflectance Components.** We evaluate different combinations of reflectance terms to understand their individual and cumulative contributions. Specifically, we compare: (A) Diffuse only, serving as a baseline; (B) adding specular; (C) adding subsurface scattering; and (D) the full model with all terms. We also examine ablations from the full model by removing: (E) the specular term, or (F) the scattering term. The full model (D) achieves the best overall performance. Subtractive ablations confirm these trends: removing either specular (E) or scattering (F) leads to noticeable degradation.

**Training Schedule.** We examine alternative strategies for introducing reflectance components during training: (H) joint training of all terms from the start; (I) our progressive schedule (Diffuse → Shadow → Scatter → Specular); (J) a non-physical variant swapping the last two; and (K) a variant that adds all terms together after a diffuse-only warm-up. The results demonstrate the effectiveness of our progressive strategy (I), which yields superior reconstruction of reflectance components.

While the quantitative results across different compositions and training schedules remain relatively close (see Tab. 2 and the additional scenes in Tab. 4), the visual decompositions (see Fig. 5 and

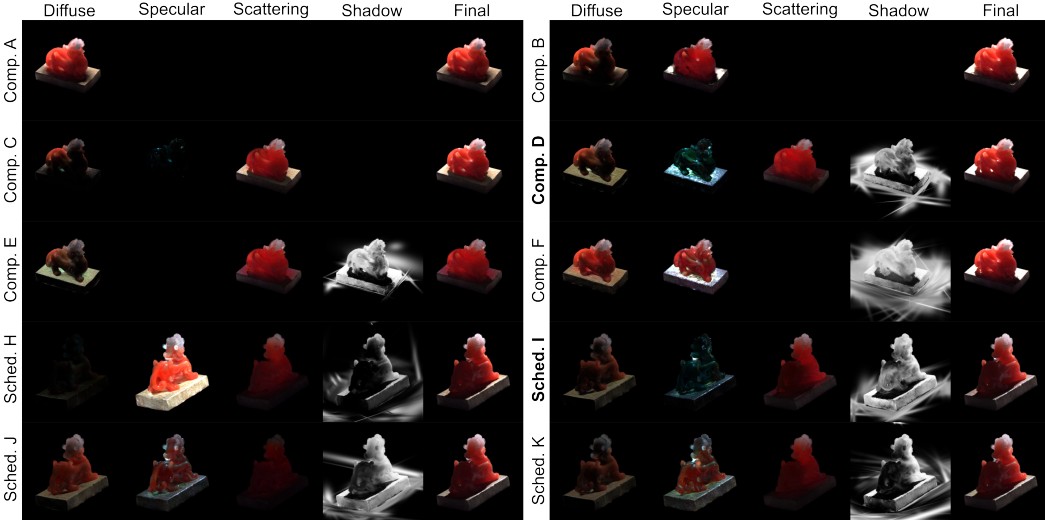

Figure 5: Visualization of reconstructed components under different reflectance decompositions and training schedules on the *Pixiu* scene from the NRHints real dataset. Top: six reflectance compositions (Comp. A-F); Bottom: four training schedules (Sched. H-K).

Table 2: Ablation study results on the real scene *Pixiu*.

| Dataset | Train Set | | | Test Set | | |
|---|---|---|---|---|---|---|
| Method | PSNR ↑ | SSIM ↑ | LPIPS ↓ | PSNR ↑ | SSIM ↑ | LPIPS ↓ |
| A: Diff | 20.2869 | 0.5701 | 0.1055 | 20.1878 | 0.5583 | 0.1061 |
| B: D + S | 20.7321 | 0.6800 | 0.0966 | 20.5692 | 0.6683 | 0.0992 |
| C: D + S + SSS | 25.1545 | 0.9336 | 0.0825 | 24.8100 | 0.9274 | 0.0857 |
| D: Full (Ours) | 33.6065 | 0.9524 | 0.0751 | 31.1213 | 0.9452 | 0.0791 |
| E: Full – S | 32.3487 | 0.9489 | 0.0817 | 30.5952 | 0.9429 | 0.0844 |
| F: Full – SSS | 31.5332 | 0.7318 | 0.0818 | 30.4095 | 0.7204 | 0.0850 |
| H: Joint | 32.5452 | 0.9500 | 0.0781 | 31.0880 | 0.9441 | 0.0812 |
| I: Prog. Phys (Ours) | 33.6065 | 0.9524 | 0.0751 | 31.1213 | 0.9452 | 0.0791 |
| J: Prog. NonPhys | 32.5606 | 0.9499 | 0.0776 | 31.0973 | 0.9443 | 0.0807 |
| K: Prog. Merge | 33.3438 | 0.9520 | 0.0758 | 31.0486 | 0.9449 | 0.0794 |

Fig. 12) show meaningful differences that highlight the importance of proper terms and training strategies. For example, in *Composition F*, removing the scattering term leads to noticeable artifacts, where both the diffuse and shadow components begin to absorb scattering, resulting in a more translucent appearance that compromises the sharpness of shadows. Similarly, in *Schedule K*, introducing multiple reflectance terms simultaneously causes training interference, where overlapping gradients between specular and scattering degrade the disentanglement quality. These artifacts are less evident in scalar metrics but manifest clearly in the visual outputs, underscoring the need for structured supervision and progressive learning.

## 6 CONCLUSION

We demonstrate that progressively introducing reflectance terms via a carefully designed training schedule enables our method to decompose scene illumination effectively and support relighting under novel lighting. Although we do not explicitly model multi-bounce global illumination, the combination of continuous volumetric visibility and the learned scattering term already captures the most perceptually important low-frequency indirect effects. While our current implementation relies on a rasterization-based pipeline, which does not fully capture physical light transport, future work could integrate ray or path tracing to improve physical realism. Incorporating additional supervision, such as multi-term losses, may further reduce role leakage and improve disentanglement of reflectance components. Material-aware grouping of Gaussians using the learned material latent space could produce a more structured representation, facilitating controllable relighting and scene editing. Overall, our work establishes a solid foundation for physically grounded, editable relightable rendering.

**Ethics Statement.** This work introduces a physically based relightable 3D reconstruction framework that recovers geometry and appearance from sparse image observations. The method aims to preserve the visual fidelity of captured scenes and does not infer personal identity or generate content beyond illumination variation. As with other reconstruction techniques, it may be applied to data containing human subjects or proprietary objects, and thus should be used in accordance with relevant consent, privacy, and intellectual property regulations.

**Reproducibility Statement.** We provide sufficient information in the main paper (Sec. 5) and the appendix (Sec. A and Sec. B) to support reproducibility, including details on the model design, training procedure, and evaluation setup. All datasets used for training and testing are described accordingly. We are happy to clarify any additional implementation details if needed.

**Acknowledgements.** This work was supported in part by the Marsden Fund Council managed by the Royal Society of New Zealand under Grant MFP-20-VUW-180, and internal research grant (Project No. 400876) from Victoria University of Wellington.

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
