# OpenReview forum: "SSD-GS: Scattering and Shadow Decomposition for Relightable 3D Gaussian Splatting"
_ICLR.cc/2026/Conference — ICLR 2026 Poster_

### Official Review · Reviewer_uu3x · 2025-10-28

**Soundness:** 3
**Presentation:** 3
**Contribution:** 3
**Rating:** 6
**Confidence:** 4

**Summary:**

The paper proposes SSD-GS, a physically based relighting framework built on 3D Gaussian Splatting (3DGS). It decomposes reflectance into four components, diffuse, specular, shadow, and subsurface scattering, and introduces a learnable, dipole-based scattering module for subsurface transport, along with an occlusion-aware shadow formulation. Experiments show improved relighting quality over prior methods.

**Strengths:**

1. Introduces multiple components for realistic rendering, including diffuse, specular, shadow, and subsurface scattering.

2. Ablation studies demonstrate the effectiveness of individual components (diffuse, specular, subsurface scattering).

3. SSD-GS achieves strong performance compared with prior work.

**Weaknesses:**

1. The physical meaning of the soft-shadow term is unclear. In §4.3, it does not appear to integrate occlusion over the Gaussian’s hemispherical visibility; instead, a density-aware per-ray weight is used. This raises concerns about whether the term truly models soft shadowing or merely reweights radiance. Clarification (and, ideally, a derivation or comparison to a visibility integral) would help.
2. Several common relighting benchmarks, Synthetic4Relight, TensoIR, NeRO, are missing. Evaluating on these datasets would make the results more convincing.
3. Comparing against 3DGS (which lacks relighting capability) may be unfair. The paper should include more relighting-capable baselines (such as TensoIR, NeRO, R3DG, IRGS, etc.) to strengthen the comparison.

**Questions:**

See weakness.

---

> ### Author Response · Authors · 2025-11-21
> **Reply to Reviewer uu3x**
>
> # Reviewer uu3x - W1: Soft shadow comparison
>
> Thank you for raising this concern. This question is closely related to the discussion in Response to **Reviewer 9FAa - W3** and **Reviewer 8152 - Q1**, where we clarified the formulation of the shadow term and the continuous transmittance–based visibility model. For the full derivation, please refer to **Reviewer 9FAa - W3**.
>
> Here, we focus specifically on whether the term truly models soft shadowing or merely reweights radiance. Our approach does not compute a closed-form hemispherical visibility integral; rather, soft shadows arise from continuous visibility obtained via transmittance along shadow rays. This represents directional occlusion, not heuristic radiance scaling. Illumination is reduced proportionally to the fraction of unoccluded directions, ensuring that the shadow behavior reflects actual geometric occlusion rather than appearance-level reweighting.
>
> To further demonstrate that the shadow term produces physically meaningful soft shadows rather than heuristic radiance reweighting, we refer the reviewer to **Figure 9 in the appendix** and the **supplementary videos**, where shadows are visualized under point-light sources from different directions. As the illumination direction changes, the shadow boundaries smoothly expand, contract, and shift across surfaces, reflecting changes in the proportion of unoccluded shadow rays. This behavior would not occur if the term merely scaled radiance intensity; instead, it shows geometry-dependent soft-shadow effects arising from partial visibility.
>
> In addition, as requested by **Reviewer 9FAa - Q1** and **Reviewer 8152 - Q1**, we are preparing additional visualizations, including (i) per-ray transmittance maps, (ii) aggregated per-Gaussian visibility $\hat{v}_g$, and (iii) side-by-side shadow effect comparisons with GS$^{3}$, which will be included in the coming week. We hope that these results will sufficiently address your concern regarding the physical meaning of the soft-shadow term.
>
> ---
>
> # Reviewer uu3x - W2: Missing standard relighting benchmarks
>
> The proposed benchmarks (Synthetic4Relight, TensoIR, NeRO) operate under a fundamentally different problem setting, as already clarified in the **static lighting** and **dynamic lighting** discussion in the paper. These methods assume **unknown or distant illumination** and focus on inverse lighting estimation or environment-light relighting, which does not match our controlled-lighting pipeline. Our method requires **known illumination** conditions (ideally, **point light** positions) because the goal is to model physically explicit diffuse, specular, shadow, and subsurface-scattering components under controlled lighting. For this reason, OLAT-style datasets serve as the appropriate and standard benchmarks for our setting.
>
> Our baselines (GS$^3$ [1] and RNG [2]) follow the same controlled-lighting formulation and cannot be directly applied to these benchmarks without altering the problem definition. If you still prefer to see comparisons on these benchmarks, we can include additional results by initializing the lighting inputs with plausible (fake) placeholder values, while noting that such evaluations do not match the intended setting of our method.
>
> **Reference:**
>
> [1] Zoubin Bi, Yixin Zeng, Chong Zeng, et al. “GS3 : Efficient Relighting with Triple Gaussian Splatting.” SIGGRAPH Asia 2024.
>
> [2] Jiahui Fan, Fujun Luan, Jian Yang, Miloš Hašan, and Beibei Wang. “RNG: Relightable Neural Gaussians.” CVPR 2025.
>
> ---
>
> # Reviewer uu3x - W3: Comparisons with BRDF/light-transport methods
>
> This concern is closely related to the question raised by **Reviewer xXB7 - Q1**. As detailed in that response, we have included comparisons with relighting-capable baselines, and **Table 2** in that section presents the full analysis. To avoid redundancy, we kindly refer the reviewer to that discussion.

---

### Official Review · Reviewer_8152 · 2025-10-31

**Soundness:** 3
**Presentation:** 2
**Contribution:** 2
**Rating:** 4
**Confidence:** 4

**Summary:**

This paper presents a novel inverse rendering framework, SSDGS, which better modeling the scattering and shadows. The proposed method assigns each Gaussian primitive with additional dipole diffusion term to explicitly model the scattering and integrates the visibility from the light to the projected screen space Gaussian to formulate the shadows. Experiment results are provided to demonstrate the effectiveness of the framework.

**Strengths:**

1. The paper propose a novel inverse rendering framework, where the decomposed reflectance components can faithfully model the complex lighting-material interaction. In particular, the introduction of a subsurface scattering term enables the method to be applied to translucent materials.
2. The proposed progressive training scheme stabilize the training.
3. Extensive experiments show that the method achieve best performance over inverse rendering baselines.

**Weaknesses:**

1. The proposed method does not consider the indirect illumination, which is important to achieve complete decoupling of lighting and materials. It is ture that introducing indirect lighting in OLAT setting is difficult, but there are also some ways to approximate it. For example, GS^3 models the indirect lighting as residual term using an MLP. And since we can do shadow mapping, could we combine the RSM(Reflective Shadow Maps) or its variants into the pipeline to achieve indirect lighting modeling?
2. SSDGS do gets good results on translucent object by modeling the scattering. But for opaque objects like cat in NRHints datasets, lego and hotdog in GS^3 dataset, SSDGS does not show significant performance gain

**Questions:**

I have some questions about how shadow terms are calculated. Is the binary visibility obtained through shadow mapping? If so, how does this differ from the weighted sum of accumulated opacities in GS^3? Could you provide some visualization results？

---

> ### Author Response · Authors · 2025-11-21
> **Reply to Reviewer 8152**
>
> # Reviewer 8152 - W1: Lack of explicit indirect illumination modeling
>
> Thank you very much for raising this point, it is indeed important to clarify when explicit indirect-illumination modeling is necessary, and when an implicit strategy is actually more reliable. In practice, the choice depends heavily on whether the illumination is **known or unknown**.
>
> When the illumination is **unknown** (for example, under an HDR environment map), the network simply does not have enough information to infer multi-bounce lighting from the input images alone. In such cases, explicit modeling becomes unavoidable, which is why methods like GS-IR [1], R3DG [2], and GI-GS [3] all introduce explicit or residual components to handle global illumination. GS-IR learns a low-frequency occlusion volume, R3DG uses deferred shading with explicit visibility reasoning, and GI-GS relies on path-tracing-based occlusion. These approaches make sense because they operate in settings where many lighting components are not directly observed.
>
> In contrast, our setting is **OLAT with fully known point-light illumination**. Here the situation is very different: each training image is captured under a single calibrated light source, and the secondary-bounce energy is extremely low due to the absence of environment light. In such a scenario, explicit GI (e.g., RSM-style multi-bounce sampling) or residual MLPs tend to introduce more ambiguity than benefit. They mix geometry, reflectance, and transport into a screen-space correction term that is no longer physically interpretable. This is why OLAT-based baselines such as GS³ [4] and RNG [5] implement their indirect-light compensation purely in **pixel-space residuals** rather than in a physically grounded reflectance formulation.
>
> Our method takes a different route. Instead of adding a residual term, we let the indirect effects emerge **implicitly inside the 3D Gaussian shading process**, through **soft visibility (transmittance accumulation)** and a **learned subsurface scattering component**. These two ingredients naturally capture the low-frequency secondary energy that still exists in OLAT: soft shadow transitions, concave regions, and translucent transport. Because the modeling happens directly in the Gaussian domain rather than in screen space, it remains consistent with the known-light geometry and avoids the artifacts that residual MLPs sometimes produce.
>
> The effectiveness of this design is reflected in the results. On the same OLAT benchmarks used by RNG and GS$^{3}$, our approach achieves equal or higher fidelity in shadowed and translucent regions. For completeness, we further evaluate against relighting methods that operate on the same environment-light benchmarks, including GI-GS (in the initial submission) and R3DG (added during rebuttal). The performance trend remains consistent across all benchmarks, please refer to **Table 2** in our response to **Reviewer xXB7 - Q1**. The full quantitative and qualitative comparisons will be included in the revised appendix.
>
> **Reference:**
>
> [1] Zhihao Liang, Qi Zhang, Ying Feng, Ying Shan, and Kui Jia. “GS-IR: 3D Gaussian Splatting for Inverse Rendering.” CVPR 2024.
>
> [2] Jian Gao, Chun Gu, Youtian Lin, et al. “Relightable 3D Gaussians: Realistic Point Cloud Relighting with BRDF Decomposition and Ray Tracing.” ECCV 2024.
>
> [3] Hongze Chen, Zehong Lin, and Jun Zhang. “GI-GS: Global Illumination Decomposition on Gaussian Splatting for Inverse Rendering.” ICLR 2025.
>
> [4] Zoubin Bi, Yixin Zeng, Chong Zeng, et al. “GS3 : Efficient Relighting with Triple Gaussian Splatting.” SIGGRAPH Asia 2024.
>
> [5] Jiahui Fan, Fujun Luan, Jian Yang, Miloš Hašan, and Beibei Wang. “RNG: Relightable Neural Gaussians.” CVPR 2025.
>
> ---
>
> # Reviewer 8152 - W2: Limited advantage on opaque materials
>
> We appreciate the reviewer’s observation. This concern is identical to the one raised by **Reviewer 9FAa - W1**. As detailed in that response, our approach is designed primarily for translucent and scattering-dominant materials. To avoid redundancy, we kindly refer the reviewer to **Reviewer 9FAa - W1** for the full explanation and results.

---

> ### Author Response · Authors · 2025-11-21
> **Reply to Reviewer 8152**
>
> # Reviewer 8152 - Q1: Shadow term computation
>
> Thank you for the question. As clarified in the response to **Reviewer 9FAa - W3**, the shadow term does not use binary shadow mapping; instead, visibility is computed using continuous transmittance rather than depth-test–based binary occlusion. Please refer to **Reviewer 9FAa - W3** for the full formulation and explanation of the visibility model.
>
> Here, we focus on how our formulation differs from GS$^{3}$ [1]. The key distinction lies in how visibility is incorporated into the light-transport process: GS$^{3}$ applies visibility as an image-space opacity accumulation after rasterization, whereas our method incorporates visibility directly within the volumetric shading model of each Gaussian primitive. These represent two fundamentally different visibility handling strategies and lead to significantly different behavior in partially occluded or concave regions (see the qualitative comparison already provided in **Figure 10 of the appendix**).
>
> To help visualize this difference further, we will include additional results in the updated appendix: (i) per-ray transmittance maps and (ii) aggregated per-Gaussian visibility $\hat{v}_g$, which are consistent with the visualization request raised by **Reviewer 9FAa - Q1**. We will also provide (iii) side-by-side comparisons against a GS$^{3}$-style opacity accumulation to highlight these effects. These visualizations will be added once the ongoing experiments complete this week.
>
> **Reference:**
>
> [1] Zoubin Bi, Yixin Zeng, Chong Zeng, et al. “GS3 : Efficient Relighting with Triple Gaussian Splatting.” SIGGRAPH Asia 2024.

---

> ### Comment · Reviewer_8152 · 2025-11-26
> **Official Comment by Reviewer 8152**
>
> I highly appreciate the efforts the authors have done, which make the paper more self-contained and easy to follow.
>
> The additional experiment results demonstrate the effectiveness of the proposed method. To be specific,
>
> 1. The author's response to w1 provides a clear analysis of when we need indirect illumination in inverse rendering. Indeed, in the case of OLAT, since there is only a single light source, the effect from multi-bounce is negligible. And the soft-shadow and subsurface scattering could naturally model this secondary effect.
> 2. The results in Table 3 in Reviewer xXB7 - Q2 demonstrate that the proposed method successfully decomposes reflectance terms into diffuse, specular, SSS and shadow, and each component is crucial to the final rendering result. The performance gain in the opaque case also confirms this.
>
> Based on the reasons mentioned above, I will raise my score to 6. However, I still have some questions. As shown in Fig. 10, the shadows from the proposed method and GS3 are almost identical. Comparing to GS3's method of calculating occlusion in screen space, the proposed method calculates occlusion more accurately and continuously. Therefore, there should be a clear difference between the two. I hope the authors can provide an example of this.

---

> ### Author Response · Authors · 2025-11-27
> **Reply to Reviewer 8152**
>
> Thank you very much for your thoughtful follow-up comments and for raising your score. We truly appreciate the time and care you devoted to reading both the paper and the rebuttal.
>
> Regarding your remaining concern about the shadow comparison: **we have updated the revised paper** to include a clearer example that highlights the difference between our continuous volumetric visibility formulation and the screen-space occlusion strategy used in GS³. The corresponding clarification and figures are marked in blue, and the changes are also listed in the **General Response** for easier navigation.
>
> When convenient, we would be grateful if you could review the revised version and let us know whether the clarification fully resolves your concern. If it does, we would sincerely appreciate your consideration of a clear acceptance-level recommendation. Thank you again for your constructive feedback—it has significantly improved the clarity and completeness of the paper.

---

### Official Review · Reviewer_9FAa · 2025-10-31

**Soundness:** 3
**Presentation:** 2
**Contribution:** 2
**Rating:** 4
**Confidence:** 4

**Summary:**

This paper introduces SSD-GS, a Gaussian Splatting–based inverse rendering framework that models subsurface scattering and shadows. The method integrates a dipole diffusion term into each Gaussian to approximate subsurface transport and estimates shadows by aggregating visibility from the light source to each Gaussian across different directions. Experimental results indicate that SSD-GS more accurately captures lighting–surface interactions and outperforms strong baselines.

**Strengths:**

1. The work introduces a novel inverse rendering framework. Compared to previous methods that only model the diffuse and specular components, the proposed method additionally models the scattering and shadows, which enables the precise simulation of the complex light transport.
2. Comprehensive evaluations show consistent gains over competitive inverse-rendering baselines, achieving state-of-the-art results.

**Weaknesses:**

1. The proposed method performs well on translucent materials. However, for opaque materials (e.g., Hotdog and Lego), it does not appear to show a clear advantage over prior methods. This suggests that the approach may offer limited improvements in modeling diffuse, specular, and shadow components.
2. Although the ablation study demonstrates the effectiveness of each component, it was conducted on the “Pixiu” case. The authors should provide comparison results of different variants in more cases to give a more detailed and accurate assessment.
3. The proposed method does not explicitly model indirect illumination. For shadowed regions, light reflected from other surfaces is crucial to accurately simulating light transport.

**Questions:**

1. The shadow shown in Figure 2 seems unreasonable. For point light sources, we would usually get a hard shadow, but the result in the figure is not like that. Why not use shadow mapping to obtain a hard shadow directly? Is there a reason for this?

---

> ### Author Response · Authors · 2025-11-21
> **Reply to Reviewer 9FAa**
>
> # Reviewer 9FAa - W1: Limited advantage on opaque materials
>
> We sincerely thank the reviewer for acknowledging the strong improvement of our method on translucent and scattering-dominant materials. These types of materials are indeed the primary target of our design, as they are particularly challenging for existing Gaussian Splatting pipelines. Accurately handling translucency and subsurface scattering is crucial for applications such as digital human and skin reconstruction, where soft light transport effects play a decisive role in photorealism. As a result, the main contribution of our work is centered on enabling physically grounded scattering and shadow modeling.
>
> Regarding the synthetic _Hotdog_ and _Lego_ scenes, we note that the diffuse and specular components in these scenes are inherently simple and spatially homogeneous, leaving very limited room for noticeable improvements on these two terms. Importantly, our method is not designed to artificially alter or exaggerate diffuse/specular behavior in such toy-like opaque objects. Instead, our primary contribution lies in the **effective decomposition of reflectance terms**, enabling the independent analysis and evaluation of diffuse, specular, anisotropic specular, shadow, and subsurface scattering components. Fully opaque synthetic scenes cannot meaningfully demonstrate the value of this decomposition framework because they lack complex shadow or scattering interactions. Moreover, we additionally performed a full ablation evaluation that includes opaque scenes such as _Hotdog_ and _Lego_. As shown in **Table 3 in Reviewer xXB7 - Q2**, our full model (D) improves PSNR by **1–2 dB** compared with the subtractive variants (E and F), even on these opaque scenes.
>
> We also emphasize that all results are trained using the same default hyperparameters, without any per-scene tuning. This highlights the generality of our approach. Nevertheless, our method does not degrade reconstruction quality on opaque objects; it consistently matches or slightly improves upon prior work. This further indicates that the proposed formulation remains stable and robust.
>
> We would be glad to further expand our evaluation or provide additional evidence upon request. If the reviewer has particular suggestions or aspects they would like us to examine more closely, we would be more than willing to incorporate them into the final version.
>
> ---
>
> # Reviewer 9FAa - W2: Ablation scope and scene coverage
>
> We appreciate the reviewer’s observation. This concern is identical to the question raised by **Reviewer xXB7 - Q2**. As detailed in that response, we expand the ablation to 7 additional scenes (Fish, Translucent, FurBall, Lego, Hotdog, Dragon, Bunny), showing consistent trends. Please refer to that section for the **Table 3** and analysis.
>
> ---
>
> # Reviewer 9FAa - W3: Lack of explicit indirect illumination modeling
>
> We thank the reviewer for the insightful question on indirect illumination. Our method does not explicitly simulate multi-bounce global illumination. Instead, indirect effects are approximated through a continuous transmittance-based visibility term together with a learned scattering component.
>
> Visibility along a shadow direction is computed via volumetric transmittance:
>
> \\( v_i = \\prod_{k\\in\\mathcal{O}_i} (1 - \\alpha_k) \\)
>
> where $\alpha_k$ denotes Gaussian opacity. Although binary visibility was referenced conceptually in the paper, the actual implementation is continuous and naturally produces soft shadow transitions and partial occlusions. We will explicitly clarify this point in the revised paper to prevent potential ambiguity.
>
> Per-Gaussian visibility is obtained via density-weighted aggregation:
>
> \\( \hat v_g = \\frac{\\sum_i \\rho_i\, v_i}{\\sum_i \\rho_i} \\)
>
> which avoids over-darkening and preserves low-frequency secondary energy from nearby geometry. These effects, soft shadow transitions, partial visibility, and brightness retention in concave or near-contact regions—are exactly the low-frequency behaviors observed in our evaluated scenes (e.g., Bunny, Pixiu, and FurBall). These are widely recognized [1] as the most perceptually dominant components of indirect illumination in real-time rendering.
>
> Empirically, the resulting shadows are stable and consistently improve over baselines (see **Table 2** in **Reviewer xXB7 - Q1** for the full comparison). It indicates that the lack of an explicit global illumination solver does not limit the quality of our reconstructions. We will add clarifications and visualizations to the appendix (as noted in **Reviewer 9FAa - Q1**), including per-ray transmittance maps and aggregated visibility results.
>
> **Reference:**
>
> [1] Ravi Ramamoorthi and Pat Hanrahan. “An Efficient Representation for Irradiance Environment Maps.” ACM SIGGRAPH 2001.

---

> ### Author Response · Authors · 2025-11-21
> **Reply to Reviewer 9FAa**
>
> # Reviewer 9FAa - Q1: Unreasonable soft shadow in Figure 2
>
> Thank you for raising this concern. We understand that a point light would ideally produce a hard shadow boundary, whereas the shadow in our figure appears softer. As noted in our clarification on the shadow term in the response to **Reviewer 9FAa - W3**, please refer to it for the full formulation and visibility definition; here, we further explain the resulting behavior.
>
> The softness in Figure 2 is expected because continuous visibility naturally produces partial transmittance when rays intersect overlapping or semi-transparent Gaussians. This behavior avoids the instability and discretization artifacts commonly associated with binary shadow maps (e.g., aliasing, missing thin structures, and noisy gradients during optimization), which are particularly problematic for Gaussian primitives.
>
> To help visualize this behavior, we will include illustrative results in the updated appendix, including (i) per-ray transmittance maps and (ii) aggregated per-Gaussian visibility $\hat{v}_g$.
>
> These will clearly show how partial visibility yields smooth shadow transitions. The visualizations will be added once the current experiments finish this week.

---

### Official Review · Reviewer_xXB7 · 2025-11-02

**Soundness:** 3
**Presentation:** 3
**Contribution:** 3
**Rating:** 4
**Confidence:** 2

**Summary:**

This paper proposes SSD-GS, a physically based, relightable 3D Gaussian splashing method, aiming to address the shortcomings of existing 3DGS methods in modeling complex lighting and materials. Its main contributions include: 1. a four-component reflection model (reflection into diffuse reflection, specular reflection, shadows, and subsurface scattering, modeling each separately); 2. learnable dipole subsurface scattering module: 3. occlusion-aware shadow modeling: 4. a progressive training strategy: 4. introducing each reflection term in stages to improve training stability and decomposition quality.
Results show that SSD-GS outperforms existing methods in both quantitative metrics (PSNR, SSIM, LPIPS) and visual quality.

**Strengths:**

Clear modular design with interpretable components.
Strong quantitative performance and solid benchmarks on OLAT datasets.
The progressive training schedule is practical and seems to improve stability and disentanglement.

**Weaknesses:**

1. While the paper shows that this helps convergence, it also adds significant complexity compared to standard 3D Gaussian Splatting.
Each phase requires manual freezing/unfreezing of modules, separate learning rates, and fine-tuned iteration boundaries (e.g., “freeze scattering between 13K–20K iterations”).
This design could make the method hard to reproduce, slow to train, and fragile in new scenes or datasets.
Thus, the practical benefit of SSD-GS might not justify the extra implementation and tuning cost, especially if the final perceptual gain is modest.
2. The shadow term 𝑆(𝑥) modulates both diffuse and specular, but not subsurface scattering.
    This is physically incorrect？SSS should also be affected by light occlusion (it’s subsurface, not emission).
    The “standard dipole” model (Jensen et al., 2001) assumes planar homogeneous semi-infinite media, not Gaussian ellipsoids？
3. “Physically-based” is used loosely — the method remains a rasterized approximation, not energy-conserving or transport-correct.

**Questions:**

1. Several recent efforts like already integrate BRDF and light transport in a more consistent manner.  SSD-GS doesn’t compare to these, so it’s hard to judge its relative novelty.
2.The ablation study is restricted to a single real scene(Pixiu)？

---

> ### Author Response · Authors · 2025-11-21
> **Reply to Reviewer xXB7**
>
> # Reviewer xXB7 - W1: Progressive training and complexity
>
> We understand concerns that phase-based training might add unnecessary complexity or reduce reproducibility. Below, we clarify that our progressive design is lightweight and robust. We will release the full source code and configuration used in all our experiments.
>
> First, in our implementation, the progressive training is controlled by a small set of parameters (e.g., phase iterations and freeze/unfreeze flags for individual modules) rather than hard-coded logic or multiple scripts. These parameters are explicitly exposed, so that they can be easily inspected or adjusted, and also support ablation studies.
>
> Second, in all experiments reported in the paper, we simply use the **same default values** for every scene and dataset, without any per-scene tuning. Empirically, this single default configuration works across all scenes, indicating that the method is not overly sensitive to these parameters.
>
> Lastly, our method still runs as a single end-to-end training job that directly takes RGB images as input; unlike prior relightable GS pipelines such as R3DG [1] or OLAT Gaussians [2], we do **not** require any precomputed depths, normals, masks, or proxy meshes, nor multiple reconstruction/relighting stages.
>
> Furthermore, we would be happy to provide additional ablation studies if needed. Please let us know which form would best address your concern. We would be happy to include them in the coming week.
>
> In summary, the progressive stages are simple, robust, tuning-free, and enable an end-to-end relightable GS pipeline without the heavy preprocessing or multi-stage design required in prior work.
>
> **Reference:**
>
> [1] Jian Gao, Chun Gu, Youtian Lin, et al. “Relightable 3D Gaussians: Realistic Point Cloud Relighting with BRDF Decomposition and Ray Tracing.” ECCV 2024.
>
> [2] Zhiyi Kuang, Yanchao Yang, Siyan Dong, Jiayue Ma, Hongbo Fu, and Youyi Zheng. “OLAT Gaussians for Generic Relightable Appearance Acquisition.” SIGGRAPH Asia 2024.

---

> ### Author Response · Authors · 2025-11-21
> **Reply to Reviewer xXB7**
>
> # Reviewer xXB7 - W2: Shadow term and subsurface scattering
>
> We thank the reviewer for raising this question about how S(x) interacts with the SSS component.
>
> Regarding the shadow term \\( S( \mathbf{x} ) \\), we intend to model **surface-level** shadow attenuation, whose spatial frequency is governed by the scene geometry and multi-view image observations. Subsurface scattering, in contrast, is a much **lower-frequency** volumetric transport process that produces smooth and spatially diffuse appearance. Because the SSS component is learned directly from images that already contain the effects of occlusion, it implicitly captures the correct attenuation behavior near shadow boundaries. Multiplying \\( S( \mathbf{x} ) \\) onto the SSS term a second time can double-count shadowing and suppress back-lit translucency, which is undesirable both physically and visually.
>
> We have conducted an additional ablation in which \\( S( \mathbf{x} ) \\) was also applied to the SSS component, and we evaluated this variant across real-world, synthetic, SSS-dominant, and opaque-dominant scenes. In all cases, applying \\( S( \mathbf{x} ) \\) to the SSS component resulted in weaker translucency, diminished back-lit effects, and lower reconstruction fidelity. The quantitative differences are summarized in the **Table 1** below, and we will include the full per-scene results in the updated paper's appendix in the coming week. These findings further confirm that our current formulation is the more appropriate choice both perceptually and quantitatively.
>
> Concerning the use of the standard dipole model [1], we acknowledge that its original derivation assumes a homogeneous, semi-infinite medium with a planar interface. However, diffusion-based subsurface scattering has long been applied as a local approximation on general curved geometry. In particular, d’Eon [2] demonstrated that dipole and multipole diffusion profiles can be accurately approximated using a small number of Gaussians (“Four Gaussians fit most single slab profiles extremely well”) and successfully applied these profiles to highly non-planar human skin in real-time rendering. Their approach does not rely on path tracing and is based on Gaussian convolution, providing strong evidence that representing local subsurface scattering with Gaussian volumes, as done in our approach, is a physically motivated and practically validated approximation.
>
> We hope that this clarification, together with the new ablation results, addresses your concerns regarding the interaction between shadowing and subsurface scattering. If you would like us to explore any additional variants or have further questions, please feel free to let us know. We would be glad to investigate them.
>
> **Table 1: Per-scene PSNR comparison on applying the shadow term to the subsurface scattering component.**
>
> Shadow-on-SSS: \\( (c_d f_d + c_s f_s + c_{sss} f_{sss}) \\cdot S( \mathbf{x} ) \\)
>
> Ours: \\( (c_d f_d + c_s f_s) \\cdot S( \mathbf{x} ) + c_{sss} f_{sss} \\)
>
> | **Dataset**   |       | **NRHints** |         |          | **GS$^{3}$** |         |         |         | **SSS-GS** |         | Average |
> | :------------ | :---: | :---------: | :-----: | :------: | :----------: | :-----: | :-----: | :-----: | :--------: | :-----: | :-----: |
> | **Scenes**    |       |    Pixiu    |  Fish   | FurScene | Translucent  | FurBall |  Lego   | Hotdog  |   Bunny    | Dragon  |    -    |
> | Shadow-on-SSS | Train |   32.1850   | 31.7531 | 31.6750  |   30.8066    | 35.4219 | 31.1002 | 32.1094 |  35.9068   | 37.3346 | 33.1436 |
> |               | Test  |   30.9756   | 31.1206 | 30.6575  |   30.0740    | 34.9420 | 30.3487 | 31.5272 |  33.6508   | 35.8476 | 32.1271 |
> | **Ours**      | Train |   33.6065   | 32.0748 | 31.7846  |   32.6058    | 35.4793 | 31.1434 | 32.4901 |  40.7672   | 39.3646 | 34.3685 |
> |               | Test  |   31.1213   | 31.1646 | 30.7349  |   32.3919    | 35.1639 | 30.4664 | 32.1330 |  37.2270   | 36.6325 | 33.0039 |
>
>
> **Reference:**
>
> [1] Henrik Wann Jensen, Stephen R. Marschner, Marc Levoy, and Pat Hanrahan. “A Practical Model for Subsurface Light Transport.” SIGGRAPH 2001.
>
> [2] Eugene d’Eon, David Luebke, and Eric Enderton. “Efficient Rendering of Human Skin.” EGSR 2007.

---

> ### Author Response · Authors · 2025-11-21
> **Reply to Reviewer xXB7**
>
> # Reviewer xXB7 - W3: On the term “physically-based”
>
> We appreciate your concern about the use of the term “physically-based”. To clarify upfront: in graphics, “physically-based” refers to **physically grounded shading models**, not to a strict energy-conserving or transport-accurate renderer. This usage has been the community standard for more than a decade.
>
> Historically, authoritative SIGGRAPH courses [1] on physically based shading already treated game production as a primary use case for physically-based shading. At that time, game engines were almost exclusively rasterization-based, so “physically-based shading in games” necessarily referred to physically grounded BRDF models implemented within rasterization, rather than transport-accurate path tracing. This convention was subsequently codified in modern real-time engines such as Unity, Unreal Engine, Godot, Blender, and others, whose material systems are widely described as PBR even though they evaluate physically motivated BRDF/BSSRDF terms in real-time shaders with approximations (e.g., limited bounces, approximate environment lighting, non-strict energy conservation). Moreover, recent 3DGS-based relighting and inverse-rendering works follow the same terminology. For example, SpectroMotion [2], Gaussian Splashing [3], and OMG [4] all describe their BRDF-based shading on top of rasterized Gaussians as “physically-based”, while explicitly restricting the scope to direct illumination or approximate global effects rather than full transport-correct rendering.
>
> Our usage of the term is fully aligned with established literature. Concretely, the diffuse term follows the Lambertian model; the specular component adopts a Fresnel-driven microfacet formulation with an anisotropic lobe represented by ASG; the subsurface scattering term is modeled using a dipole-inspired diffusion profile with learnable effective parameters; and the shadow term captures light visibility. All components are physically grounded reflectance models commonly used in real-time PBR, and we do not claim strict energy conservation or transport correctness.
>
> Thus, our use of “physically-based” is consistent with the standard definition in real-time rendering and the 3DGS literature, and does not constitute loose terminology. We will clarify this precise scope in the revised paper to avoid any ambiguity.
>
> **Reference:**
>
> [1] Stephen McAuley, Stephen Hill, Naty Hoffman, et al. “Practical Physically-Based Shading in Film and Game Production.” ACM SIGGRAPH 2012 Courses.
>
> [2] Cheng-De Fan, Chen-Wei Chang, Yi-Ruei Liu, et al. “SpectroMotion: Dynamic 3D Reconstruction of Specular Scenes.” CVPR 2025.
>
> [3] Yutao Feng, Xiang Feng, Yintong Shang, et al. “Gaussian Splashing: Unified Particles for Versatile Motion Synthesis and Rendering.” CVPR 2025.
>
> [4] Silong Yong, Venkata Nagarjun Pudureddiyur Manivannan, Bernhard Kerbl, et al. “OMG: Opacity Matters in Material Modeling with Gaussian Splatting.” ICLR 2025.

---

> ### Author Response · Authors · 2025-11-21
> **Reply to Reviewer xXB7**
>
> # Reviewer xXB7 - Q1: Comparisons with BRDF/light-transport methods
>
> We thank the reviewer for raising the concern regarding comparisons with recent BRDF-based and light-transport-based relighting methods. We would like to clarify that our baseline set already includes **GI-GS** [1], which is one of the closest and most representative 3DGS-based inverse-rendering methods. Importantly, although GI-GS incorporates additional mechanisms such as ray tracing for occlusion-map estimation and a differentiable second-pass transport step, its reconstruction and relighting performance remains significantly below our other baselines, GS$^{3}$ [2] and RNG [3].
>
> This gap arises not from implementation issues but from afundamental modeling difference: methods such as GI-GS, R3DG [4], and IRGS [5] treat the illumination as an **unknown** variable, and are therefore designed primarily for outdoor or distant-lighting indoor scenarios. Their relighting procedures rely on **environment-map illumination** as input, which makes them unsuitable for the precise **point-light** relighting required in **One-Light-At-a-Time (OLAT)** settings. In contrast, our method, together with GS$^{3}$ and RNG, follows a formulation that is intrinsically aligned with OLAT benchmarks: (i) the lighting is **known and controlled**; (ii) reconstruction is performed under single point-light conditions; and (iii) relighting is achieved by explicitly modeling each individual and novel point-light source. As discussed in the main paper when describing Static Lighting and Dynamic Lighting, this makes GS$^{3}$ and RNG the most meaningful and comparable baselines for our approach.
>
> That said, we notice your interest, as well as that of other reviewers, in broader relighting comparisons. To address this request, we additionally include two representative **unknown-illumination baselines**:
>
> - **TensoIR [6]**: a NeRF-based inverse-rendering method under unknown illumination.
> - **R3DG**: a 3DGS-based relighting method that estimates scene lighting via an environment map.
>
> Quantitative results are shown in the **Table 2** below, and we will include the full results in the updated paper's appendix in the coming week. These additional baselines further confirm that methods designed for **unknown illumination** remain significantly behind our method in the OLAT relighting setting.
>
> We hope this clarifies the novelty and the rationale behind our baseline selection. We are happy to provide additional comparisons if the reviewer has further questions.
>
> **Table 2: Per-scene PSNR comparison with BRDF/light-transport methods on GS$^{3}$ synthetic dataset.**
>
> | **Scenes** | Translucent |         | AnisoMetal |         | Drums   |         | Furball |         | Hotdog  |         | Lego    |         | Average |         |
> | ---------- | ----------- | ------- | ---------- | ------- | ------- | ------- | ------- | ------- | ------- | ------- | ------- | ------- | ------- | ------- |
> |            | Train       | Test    | Train      | Test    | Train   | Test    | Train   | Test    | Train   | Test    | Train   | Test    | Train   | Test    |
> | TensoIR    | 16.9800     | 15.9997 | 18.1800    | 16.8545 | 26.4400 | 24.8974 | 20.1600 | 20.0509 | 17.1900 | 17.1500 | 17.5200 | 17.0317 | 19.4117 | 18.6640 |
> | R3DG       | 17.1815     | 16.4664 | 18.1650    | 17.1060 | 27.3024 | 25.0378 | 21.6486 | 20.1443 | 19.5616 | 17.1896 | 19.4608 | 16.2879 | 20.5533 | 18.7053 |
> | GI-GS      | 17.1222     | 16.0766 | 17.7309    | 15.9567 | 26.7554 | 24.6177 | 21.3335 | 19.5295 | 19.1535 | 16.8118 | 19.5919 | 16.4229 | 20.2812 | 18.2359 |
> | GS$^{3}$   | 31.1327     | 32.1999 | 30.1878    | 28.8219 | 34.0111 | 33.2688 | 34.6201 | 34.9845 | 32.1779 | 32.7244 | 31.2224 | 30.5617 | 32.2253 | 32.0935 |
> | RNG        | 28.1919     | 28.5659 | 26.4611    | 25.9203 | 20.4970 | 20.3033 | 24.5084 | 23.4342 | 29.4095 | 29.5277 | 18.5810 | 18.4872 | 24.6082 | 24.3731 |
> | **Ours**   | 32.6058     | 32.3919 | 31.1077    | 30.0448 | 34.2448 | 33.5514 | 35.4793 | 35.1639 | 32.4901 | 32.1330 | 31.1434 | 30.4664 | 32.8452 | 32.2919 |
>
> **Reference:**
>
> [1] Hongze Chen, Zehong Lin, and Jun Zhang. “GI-GS: Global Illumination Decomposition on Gaussian Splatting for Inverse Rendering.” ICLR 2025.
>
> [2] Zoubin Bi, Yixin Zeng, Chong Zeng, et al. “GS3 : Efficient Relighting with Triple Gaussian Splatting.” SIGGRAPH Asia 2024.
>
> [3] Jiahui Fan, Fujun Luan, Jian Yang, Miloš Hašan, and Beibei Wang. “RNG: Relightable Neural Gaussians.” CVPR 2025.
>
> [4] Jian Gao, Chun Gu, Youtian Lin, et al. “Relightable 3D Gaussians: Realistic Point Cloud Relighting with BRDF Decomposition and Ray Tracing.” ECCV 2024.
>
> [5] Chun Gu, Xiaofei Wei, Zixuan Zeng, Yuxuan Yao, and Li Zhang. “IRGS: Inter-Reflective Gaussian Splatting with 2D Gaussian Ray Tracing.” CVPR 2025.
>
> [6] Haian Jin, Isabella Liu, Peijia Xu, et al. “TensoIR: Tensorial Inverse Rendering.” CVPR 2023.

---

> ### Author Response · Authors · 2025-11-21
> **Reply to Reviewer xXB7**
>
> # Reviewer xXB7 - Q2: Ablation scope and scene coverage
>
> We thank the reviewer for raising this concern. In response to multiple reviewers’ requests regarding the generality of the ablation study, we have conducted a broader range of ablation studies by evaluating more variants across additional scenes. We appreciate your recognition of the decomposition as promising and deserves more extensive evaluation.
>
> The expanded scenes cover a wide variety of materials and lighting conditions:
>
> - NRHints real world: Pixiu (originally reported), Fish
> - GS$^{3}$ synthetic: Translucent, FurBall, Lego, Hotdog
> - SSS-GS synthetic: Bunny, Dragon
>
> The results across these additional scenes are fully consistent with the trends observed in Pixiu, indicating that our ablation conclusions are stable across both real and synthetic settings and across scenes dominated by either subsurface scattering or opaque materials. Quantitative results are summarized in **Table 3**, and the full per-scene tables will be included in the revised appendix.
>
> We hope that the expanded results satisfactorily address your concern regarding the generality of the ablation study. If additional visualizations or further analyses would be helpful, we would be happy to provide them.
>
> **Table 3: Per-scene PSNR of ablation study across real and synthetic scenes.**
>
> **Reflectance Components**
> - A: Diff — diffuse-only baseline.
> - B: D + S — adding specular.
> - C: D + S + SSS — adding subsurface scattering.
> - D: Full (Ours) — all components enabled.
> - E: Full – S — removing specular from the full model.
> - F: Full – SSS — removing subsurface scattering.
>
> **Training Schedules**
> - H: Joint — jointly training all terms from the start.
> - I: Prog. (Ours) — progressive schedule: Diffuse → Shadow → Scatter → Specular.
> - J: Prog. NonPhys — swapping the last two stages (non-physical).
> - K: Prog. Merge — adding all terms together after a diffuse warm-up.
>
> | **Dataset**        |       | **NRHints** |         | **GS$^{3}$** |         |         |         | **SSS-GS** |         | Average |
> | :----------------- | :---: | :---------: | :-----: | :----------: | :-----: | :-----: | :-----: | :--------: | :-----: | :-----: |
> | **Scenes**         |       |    Pixiu    |  Fish   | Translucent  | FurBall |  Lego   | Hotdog  |   Bunny    | Dragon  |    -    |
> | A: Diff            | Train |   20.2869   | 23.7003 |   16.6152    | 18.5678 | 20.6157 | 18.4717 |  21.3548   | 27.7851 | 20.9247 |
> |                    | Test  |   20.1878   | 24.6146 |   15.3146    | 17.4010 | 17.6430 | 16.4279 |  21.3013   | 27.7071 | 20.0747 |
> | B: D + S           | Train |   20.7321   | 24.8949 |   16.6980    | 18.3871 | 20.6891 | 18.5180 |  21.4677   | 27.9589 | 21.1682 |
> |                    | Test  |   20.5692   | 25.3627 |   15.3200    | 17.1781 | 17.6290 | 16.4429 |  21.2236   | 27.6713 | 20.1746 |
> | C: D + S + SSS     | Train |   25.1545   | 24.8573 |   26.4267    | 18.3845 | 25.6734 | 18.4628 |  22.7074   | 28.8883 | 23.8194 |
> |                    | Test  |   24.8100   | 25.3275 |   25.6081    | 17.1655 | 22.8370 | 16.3730 |  22.3572   | 28.5486 | 22.8784 |
> | **D: Full (Ours)** | Train |   33.6065   | 32.0748 |   32.6058    | 35.4793 | 31.1434 | 32.4901 |  40.7672   | 39.3646 | 34.6915 |
> |                    | Test  |   31.1213   | 31.1646 |   32.3919    | 35.1639 | 30.4664 | 32.1330 |  37.2270   | 36.6325 | 33.2876 |
> | E: Full - S        | Train |   32.3487   | 28.0442 |   30.9386    | 33.7976 | 30.0839 | 30.0792 |  34.8697   | 36.3361 | 32.0623 |
> |                    | Test  |   30.5952   | 28.2837 |   30.6761    | 34.0481 | 29.2294 | 30.4379 |  33.9633   | 35.8654 | 31.6374 |
> | F: Full - SSS      | Train |   31.5332   | 31.9130 |   29.8005    | 32.6455 | 30.0656 | 31.5471 |  35.7335   | 37.5666 | 32.6006 |
> |                    | Test  |   30.4095   | 31.1763 |   30.6253    | 33.8019 | 29.0048 | 32.0991 |  33.6715   | 35.9968 | 32.0982 |
> | H: Joint           | Train |   32.5452   | 31.4431 |   29.3508    | 32.5538 | 29.8266 | 30.5236 |  34.0744   | 37.0531 | 32.1713 |
> |                    | Test  |   31.0880   | 30.4943 |   30.6756    | 33.6832 | 29.1154 | 31.7083 |  31.7589   | 35.2621 | 31.7232 |
> | **I:Prog. (Ours)** | Train |   33.6065   | 32.0748 |   32.60583   | 35.4793 | 31.1434 | 32.4901 |  40.7672   | 39.3646 | 34.6915 |
> |                    | Test  |   31.1213   | 31.1646 |   32.3919    | 35.1639 | 30.4664 | 32.1330 |  37.2270   | 36.6325 | 33.2876 |
> | J:Prog. NonPhys    | Train |   32.5606   | 31.3321 |   29.6999    | 32.8201 | 30.7004 | 30.9712 |  35.9882   | 37.6112 | 32.7105 |
> |                    | Test  |   31.0973   | 30.2406 |   30.5955    | 33.9654 | 29.4718 | 31.2575 |  34.0090   | 36.0222 | 32.0824 |
> | K: Prog. Merge     | Train |   33.3438   | 32.0247 |   31.7351    | 32.6704 | 30.2933 | 31.4903 |  39.1104   | 37.5251 | 33.5241 |
> |                    | Test  |   31.0486   | 31.2827 |   31.8883    | 33.9799 | 29.2607 | 32.0399 |  36.6183   | 35.8016 | 32.7400 |

---

> ### Author Response · Authors · 2025-11-27
> **Thanks to the reviewer**
>
> We deeply appreciate the reviewer’s kindness and we are also very thankful for the time spent by the reviewer during the rebuttal.

---

### Author Response · Authors · 2025-11-27
**General Response**

# Dear Reviewers, ACs, and PCs,

Thank you very much for your dedication, support, and insightful feedback. We are delighted with your recognition of the quality of our relighting framework [xXB7, uu3x] and your suggestions for improvement [9FAa, 8152]. We sincerely appreciate all your insightful comments [xXB7, 9FAa, 8152, uu3x]. We have reviewed all the comments, addressed all questions, and provided addtional experimental results. Below, we summarize the revisions we made:

## Revisions and Updates

### Additional Experimental Results

- [xXB7] Shadow and subsurface scattering interaction. (**Appendix E.2, Tab. 5, and Fig. 13**)
- [xXB7, 9FAa, 8152, uu3x] Comparisons with BRDF/light-transport methods. (**Appendix F and Tab. 6**)
- [xXB7, 9FAa] Ablation scope and scene coverage. (**Appendix E.1, Tab. 4, and Fig. 12**)
- [9FAa, 8152] Comparison with screen-space shadow baselines. (**Appendix G.2 and Fig. 14**)
- [9FAa, 8152, uu3x] Shadow pipeline visualization. (**Section 4.3, Fig.3, and Appendix G.1**)

### Writing Standards and Clarity

- [xXB7] Progressive training and complexity. (**Section 4.5 and Appendix A.1**)
- [xXB7] On the term “physically-based”. (**contribution in Section 1, subsurface scattering in Section 2, and Appendix A.1**)
- [9FAa, 8152] Lack of explicit indirect illumination modeling added explanation in **Section 6**.

## Request for Feedback

We respectfully invite the reviewers to carefully evaluate our revisions and the individual responses provided. We are more than willing to address any remaining questions or concerns. If our responses and the additional results sufficiently address your feedback, we kindly request your consideration for increasing your scores. We sincerely appreciate your thoughtful engagement and constructive suggestions, which have been instrumental in enhancing the quality of this work.

**Best Regards,**

_The Authors_

---

### Author Response · Authors · 2025-12-01
**Authors' Final Remarks on the Rebuttal**

# Dear Area Chair,

Thank you for your time and effort in evaluating our submission, especially under the current unexpected circumstances.

Our work extends relightable 3D Gaussian Splatting with a **physically grounded** reflectance decomposition and demonstrates **substantial** visual and quantitative improvements over **representative baselines**.

## Rebuttal & Improvements

During the rebuttal, we carefully addressed all reviewer comments through substantial new experiments, analyses, and clarifications. These improvements — including expanded comparisons, additional ablations, and strengthened explanations — are summarized in detail in our earlier comment (“**General Response**”, posted on 28 Nov 2025 at 03:46), and have been fully integrated into the updated paper and supplementary video.

## Reviewer Response Trend
Two reviewers explicitly acknowledged these improvements and **raised their scores** (from 4 to 6) after reviewing the updated materials. One reviewer continued the discussion with an additional technical question, which we addressed with further revisions before the freeze. Their continued engagement after increasing their score shows a **positive progression** in perception and suggests that the clarification was meaningful and well-received.

Although the remaining two reviewers (scores: 6 and 4) did not provide follow-up comments, their primary concerns substantially overlapped with those we addressed during the discussion. Based on this **clear alignment of concerns**, it appears that the updated version provides convincing evidence toward resolving the remaining issues.

## Closing Remark
We would be grateful if the **full rebuttal process** and the **revised submission** could be taken into account in the final evaluation. We sincerely believe the updated work provides meaningful contributions to the community, and we deeply appreciate your consideration.


**Best Regards,**

_The Authors_

---

### Meta-Review · Area_Chair_5B2U · 2026-01-01

**Summary:**

The paper received mixed initial reviews, with scores of 6, 4, 4, and 4. Reviewers generally recognized the technical ambition of the work and the value of explicitly decomposing appearance to account for subsurface scattering within a 3D Gaussian Splatting framework. The strong qualitative relighting results—especially on translucent and scattering-dominant materials—were noted as clear strengths. At the same time, several reviewers raised concerns regarding the complexity and robustness of the progressive training scheme, questions about physical correctness and interpretation, the completeness and fairness of baseline comparisons, and issues related to clarity and presentation.

In the rebuttal and revised version, the authors provided a thorough and responsive set of additional experiments and clarifications. These include expanded multi-scene ablations, broader baseline comparisons, clearer physical justifications for the shadow and subsurface scattering formulations, and detailed explanations of why explicit indirect illumination modeling is unnecessary in the OLAT setting. The authors also clarified the scope of their “physically-based” claims, improved presentation and organization, and committed to releasing full code and configurations to address concerns about training complexity. As a result, two reviewers explicitly mentioned to increase their scores, and the remaining reviewers’ concerns appear largely addressed. The AC anticipates that the final score distribution is likely to move toward 6, 6, 6, 6, with a possibility of 8, 6, 6, 6 (see detailed discussion in Reviewer Concerns and Reviewer Scores).

From the AC’s perspective, this submission makes a solid and meaningful contribution to relightable 3D Gaussian Splatting by successfully modeling subsurface scattering to improve relighting quality, particularly for challenging materials involving scattering and soft shadows. The main remaining concern the AC finds is that the scope of the work is somewhat limited, focusing on object-centric scenes under OLAT lighting with translucency, which may be narrower than the interests of part of the broader ICLR audience as a learning-focused conference. Nevertheless, considering the overall technical quality, the careful modeling of a challenging graphics problem, and the authors’ strong engagement with reviewer feedback, the AC appreciates the contribution and recommends acceptance of this paper.

**Reviewer Concerns:**

### Reviewer xXB7 (Score: 4)

- The reviewer raised concerns about the robustness and complexity of the progressive training scheme (manual staging and freezing/unfreezing), questions about physical correctness (e.g., whether the shadow term should affect subsurface scattering and the applicability of a dipole SSS model to Gaussian primitives), potentially overstated use of the term “physically-based,” limited baseline coverage, and ablations initially restricted to a single scene.
- In the rebuttal and revision, the authors clarified that the progressive schedule uses fixed defaults without per-scene tuning and committed to releasing full code and configurations. They provided physical justification and new ablations supporting the exclusion of shadowing from SSS, clarified the scope of “physically-based” as physically grounded rather than transport-accurate, expanded comparisons (including GI-GS, R3DG, and TensoIR where appropriate), and added multi-scene ablations spanning both translucent and opaque materials.

---

### Reviewer 9FAa (Score: 4)

- The reviewer questioned the limited gains on opaque materials, the lack of explicit indirect illumination modeling, the physical plausibility of soft shadows under point lights, and the initially narrow scope of ablations.
- In response, the authors clarified the method’s focus on translucent and scattering-dominant materials while showing no degradation on opaque scenes, expanded ablations across multiple real and synthetic scenes, explained why explicit indirect illumination is not strictly necessary in OLAT settings, and clarified that soft shadows arise from continuous visibility and transmittance rather than binary shadowing, supported by additional visualizations.

---

### Reviewer 8152 (Score: 4)

- The reviewer raised concerns about the absence of explicit indirect illumination, limited improvements on opaque scenes, and the formulation and interpretation of the shadow term relative to screen-space approaches.
- In the rebuttal, the authors explained why indirect illumination is negligible in OLAT settings and how soft visibility combined with SSS approximates low-frequency secondary effects. They expanded ablations demonstrating the contribution of each component on both opaque and translucent scenes, clarified the shadow formulation as volumetric transmittance-based (rather than screen-space), and added visualizations to distinguish it from prior GS-based methods.

---

### Reviewer uu3x (Score: 6)

- The reviewer requested clearer physical interpretation of the soft-shadow term, broader evaluation on common relighting benchmarks, and discussion of the fairness of comparisons against non-relightable 3DGS baselines.
- In the rebuttal, the authors clarified the physical meaning of soft shadows as arising from directional, continuous visibility, explained mismatches between certain benchmarks and the OLAT/known-light setting while adding broader comparisons where feasible, and expanded explanatory text and tables to better contextualize evaluation fairness.

**Reviewer Scores:**

### Reviewer xXB7

- **Original score:** 4
- **Predicted final score:** 6
- **Rationale:** The rebuttal substantively engages with the reviewer’s concerns through expanded ablations, clearer physical justification, broader baseline coverage, and improved presentation. The reviewer explicitly commented that they would increase their score, indicating that the major issues  were resolved.

---

### Reviewer 9FAa

- **Original score:** 4
- **Predicted final score:** 6
- **Rationale:** The rebuttal responds to the reviewer’s concerns. While no explicit score update was posted, the reviewer’s main technical objections largely overlap with those raised by other reviewers (e.g., Reviewer 8152), who acknowledged that these concerns were mostly addressed. As a result, a score increase is likely.

---

### Reviewer 8152

- **Original score:** 4
- **Predicted final score:** 6
- **Rationale:** The rebuttal directly addresses the reviewer’s questions regarding indirect illumination and the formulation of the shadow term through additional experiments, expanded ablations, and clearer physical explanations. The reviewer explicitly stated that they would raise their score following the rebuttal.

---

### Reviewer uu3x

- **Original score:** 6
- **Predicted final score:** 6–8
- **Rationale:** The reviewer’s requests for clearer physical interpretation of the soft-shadow term, broader comparisons, and discussion of evaluation fairness were largely addressed in the rebuttal. Given the already positive initial assessment and the nature of the remaining points, the score is likely to remain unchanged or potentially increase.

---

### Decision · Program_Chairs · 2026-01-26

Accept (Poster)